# A Facile Method for Preparing UiO-66 Encapsulated Ru Catalyst and its Application in Plasma-Assisted CO_2_ Methanation

**DOI:** 10.3390/nano9101432

**Published:** 2019-10-10

**Authors:** Weiwei Xu, Mengyue Dong, Lanbo Di, Xiuling Zhang

**Affiliations:** College of Physical Science and Technology, Dalian University, Dalian 116622, China; xuweiwei0626@sina.com (W.X.); dmy_up@sina.com (M.D.)

**Keywords:** metal-organic frameworks, Ru NPs, encapsulation, CO_2_ methanation, plasma

## Abstract

With increasing applications of metal-organic frameworks (MOFs) in the field of gas separation and catalysis, the preparation and performance research of encapsulating metal nanoparticles (NPs) into MOFs (M@MOF) have attracted extensive attention recently. Herein, an Ru@UiO-66 catalyst is prepared by a one-step method. Ru NPs are encapsulated in situ in the UiO-66 skeleton structure during the synthesis of UiO-66 metal-organic framework via a solvothermal method, and its catalytic activity for CO_2_ methanation with the synergy of cold plasma is studied. The crystallinity and structural integrity of UiO-66 is maintained after encapsulating Ru NPs according to the X-ray diffraction (XRD), Fourier transform infrared spectroscopy (FTIR), and scanning electron microscopy (SEM). As illustrated by X-ray photoelectron spectroscopy (XPS), high resolution transmission electron microscopy (HRTEM), and mapping analysis, the Ru species of the hydration ruthenium trichloride precursor are reduced to metallic Ru NPs without additional reducing processes during the synthesis of Ru@UiO-66, and the Ru NPs are uniformly distributed inside the Ru@UiO-66. Thermogravimetric analysis (TGA) and N_2_ sorption analysis show that the specific surface area and thermal stability of Ru@UiO-66 decrease slightly compared with that of UiO-66 and was ascribed to the encapsulation of Ru NPs in the UiO-66 skeleton. The results of plasma-assisted catalytic CO_2_ methanation indicate that Ru@UiO-66 exhibits excellent catalytic activity. CO_2_ conversion and CH_4_ selectivity over Ru@UiO-66 reached 72.2% and 95.4% under 13.0 W of discharge power and a 30 mL·min^−1^ gas flow rate ($V_{H_{2}}:V_{CO_{2}}=4:1$), respectively. Both values are significantly higher than pure UiO-66 with plasma and Ru/Al_2_O_3_ with plasma. The enhanced performance of Ru@UiO-66 is attributed to its unique framework structure and excellent dispersion of Ru NPs.

## 1. Introduction

Metal-organic frameworks (MOFs) are a series of porous crystal materials self-assembled by metal ions and organic ligands through coordination bonds [1,2,3]. The properties of specific surface area, porosity, and tunable functional structure make MOFs promising candidates in many applications such as gas storage and separation, drug delivery, chemical sensing, and catalysis [4,5,6,7,8]. As a typical Zr-based MOF, UiO-66 was first synthesized by Cavka et al. [9] and named after the University of Oslo. It has a perfect Zr_6_O_4_(OH)_4_ octahedral framework structure and exhibits exceptional thermal stability. In addition, UiO-66 has been widely utilized in the separation and conversion of CO_2_ [6,10,11,12,13].

The utility of UiO-66 as a support for metal catalysts (M/UiO-66, M=Cu, Au, Pd, Pt, Ru, etc.) has been extensively studied by researchers recently [14,15,16,17,18,19]. For example, Milet et al. [18] reported Pt/UiO-66 catalysts prepared by the double solvent method: impregnation of the UiO-66 support with the aqueous solution of H_2_PtCl_6_, and then reduction with NaBH_4_ solution. The Pt/UiO-66 exhibited excellent performance for CO_2_ methanation. The CO_2_ conversion and CH_4_ selectivity were as high as 50% and 36% at 350 °C with a CO_2_:H_2_ molar ratio of 1:5.2 and 1650 h^−1^ gas hourly space velocity (GHSV), respectively. Compared with the simple impregnation method, encapsulating metal nanoparticles (MNPs) into the skeleton structure of the MOF template can adjust the size distribution of MNPs and prepare high-performance MNPs@MOF catalysts [20]. Li et al. [21] synthesized UiO-66-encapsulated nano-palladium catalysts (Pd@UiO-66). As described, the Pd(acac)_2_ was first prepared, and then 1 g of UiO-66 and Pd(acac)_2_ solutions were mixed to synthesize Pd@UiO-66. Small Pd NPs (2.2 nm) were obtained due to the confinement of the small pore structure of UiO-66. The synthesized Pd@UiO-66 exhibited high catalytic activity and stability for continuous catalytic upgrading of ethanol to n-butanol. The ethanol conversion and the n-butanol selectivity over the best Pd@UiO-66 catalyst was 49.9% and 50.1%, respectively, during a 200-h evaluation. The high performance was attributed to the close synergy of highly distributed Pd NPs and coordinatively unsaturated Zr sites in UiO-66. Dong et al. [22] encapsulated Pd NPs in UiO-66 with a microwave-assisted method. The pores of UiO-66 were activated, and the metal precursors were reduced at the same time in the presence of NaBH_4_. The obtained Pd@UiO-66 exhibited high catalytic activity for Suzuki–Miyaura coupling reactions at mild conditions. Therefore, using MOFs with tunable porous structures as supports, growth of the metal NPs could be confined due to the encapsulation [23]. Consequently, high-performance MOFs supported metal catalysts with small size and high dispersion metal NPs can be generally obtained [24,25,26]. However, to synthesize the above MOFs-supported metal catalysts, there are two or more steps required, the synthesis of the MOF support and the impregnation, and the reduction of the supported/encapsulated metal precursors in the presence of reducing agents [18]. Therefore, this process is generally sophisticated and time-consuming.

The CO_2_ methanation has great prospects in economic and environmental applications since most of the fuel resources and one-carbon molecules (C1) can be regenerated from CO_2_ [27,28]. The emerging plasma-assisted activation of CO_2_ for methanation can provide the high energy for CO_2_ decomposition and overcome the relatively harsh conditions and reaction devices required for conventional thermochemical conversion [29,30,31,32,33]. Ru-based catalysts, due to their efficient activity, have been applied in CO_2_ methanation extensively. This is because, in the process of CO_2_ hydrogenation, the active Ru sites are more selective to the formation of methane, which can promote the reaction of CO_2_ along the methanation path in the hydrogen-rich environment [34,35,36,37].

In this work, we report a simple and efficient method for synthesizing Ru@UiO-66 via in situ encapsulation of Ru NPs by the reduction of the RuCl_3_ precursor during the growth of UiO-66 framework structure. The crystallinity and structural integrity of Ru@UiO-66 are similar to UiO-66. The synthesized Ru@UiO-66 exhibits high performance for plasma-assisted catalytic CO_2_ methanation.

## 2. Experimental

### 2.1. Materials

Zirconium tetrachloride (ZrCl_4_, 98%), terephthalic acid (BDC, 99%) and hydrochloric acid (HCl, 37%) were supplied by Sinopharm Chemical Reagent Co., Ltd. (Shanghai, China). *N,N*-dimethylformamide (DMF, 99%) and hydration ruthenium trichloride (RuCl_3_·xH_2_O, Ru content 37%) were purchased from Tianjin Zhiyuan Co., Ltd. (Tianjin, China) and Walixi Chemical Co., Ltd. (Guangdong, China), respectively. Anhydrous methanol (CH_3_OH, 99.5%) was bought from Tianjin Kermel Co., Ltd. (Tianjin, China). All chemicals were used without further purification. High purity H_2_ (>99.999%) was generated by an HGH-500E hydrogen generator, and high purity CO_2_ and Ar (>99.999%) were obtained from the Guangming Research & Design Institute of Chemical Industry Co., Ltd. (Dalian, China).

### 2.2. Synthesis of UiO-66 and Ru@UiO-66

The strategy for preparing UiO-66 and Ru@UiO-66 is depicted in Figure 1. UiO-66 was prepared by a solvent thermal method previously reported in [38]. In brief, 1.165 g of ZrCl_4_ and 0.831 g of BDC were dissolved in 30 mL DMF, and then 0.8 mL of concentrated HCl (37%) was added. The obtained mixture was placed in an ultrasonic reactor for 20 min and heated at 120 °C for 24 h in a Teflon-lined steel autoclave. After cooling to room temperature, the obtained suspension of UiO-66 was centrifuged by a centrifuge machine (9000 rpm, 10 min), and rinsed with 25 mL of DMF and 25 mL of anhydrous methanol three times, respectively. The obtained UiO-66 was dried at 100 °C under vacuum. The synthesis procedure of Ru@UiO-66 was similar to that of UiO-66. The difference was that 0.415 g of RuCl_3_ with 1.165 g of ZrCl_4_ and 0.831 g of BDC were added into 30 mL of DMF. The other parameters were the same as that for synthesizing the pure UiO-66. The mass fraction of Ru determined by inductively coupled plasma optical emission spectrometer (ICP-OES) in Ru@UiO-66 was 2.83 wt%.

### 2.3. Catalytic Evaluation

The catalytic activity of the UiO-66 supported Ru catalysts was evaluated via a dielectric barrier discharge (DBD) plasma-assisted catalytic CO_2_ methanation reaction. The DBD plasma reactor is composed of a coaxial quartz tube (inner diameter: 8 mm, outer diameter: 10 mm), copper rod inner electrode (diameter: 2 mm) and 1 mm of a copper coil ground electrode. The discharge length and discharge gap are 25 mm and 2.5 mm, respectively. Typically, 0.3 g of samples were placed in the discharge area at a sinusoidal peak-to-peak voltage of 19.2 kV. The working gases of CO_2_ and H_2_ were mixed into the reactor after being measured by the mass flow meter (total flow rate was 30 mL·min^−1^, *V*_H__2_:*V*_CO__2_ = 4:1), while the outlet gas flow rate was measured by a soap film bubble flow meter. The gaseous products were analyzed online by a gas chromatograph (Tianmei GC-7890, Shanghai, China) equipped with a thermal conductivity detector (TCD). A type of FLUKE MT4 Max+ IR thermometer was used to monitor the temperature during the discharge process. The temperature of the reaction was ca. 200 °C. To evaluate the performance of CO_2_ methanation under cold plasma-assistance, CO_2_ conversion (*X_CO_*_2_), selectivity (*S*), and yield (*Y*) of the products were calculated as the reference [39]:(1)$$X_{CO_{2}}=\frac{F_{CO_{2}}-F_{CO_{2}}^{\prime }}{F_{CO_{2}}}\times 100\%$$
(2)$$S_{CO}=\frac{F_{CO}^{\prime }}{F_{CO_{2}}-F_{CO_{2}}^{\prime }}\times 100\%$$
(3)$$S_{CH_{4}}=\frac{F_{CH_{4}}^{\prime }}{F_{CO_{2}}-F_{CO_{2}}^{\prime }}\times 100\%$$
(4)Y=X×S×100%
where *F* and *F′* are the inlet and outlet gas flow rates, respectively. 

### 2.4. Catalysts Characterization

The X-ray diffraction (XRD) patterns were completed using a DX-2700 (Dandong, China) diffractometer with Cu Kα radiation (*λ* = 1.54178 Å) at 40 kV and 30 mA, and the step size of the measurement was 0.03°. Fourier transform infrared (FTIR) spectra were recorded in the range of 400–4000 cm^−1^ on a Nicolet AVATAR 370 (Waltham, MA, USA) infrared spectrometer. A Zeiss Sigma 500 (Jena, Germany) scanning electron microscope (SEM), operating at 10 kV, was utilized to characterize the morphology of the sample crystals. The X-ray photoelectron spectroscopy (XPS) spectra were recorded on a Thermo ESCALAN 250 spectrometer (Waltham, MA, USA) with Al Kα (1486.6 eV). The high-resolution transmission electron microscopy (HRTEM) and mapping were measured on a Tecnai G2 f20 s-twin (Hillsboro, OR, USA) transmission electron microscope. Binding energies were calibrated using C1s (284.6 eV) as the standard. The spectra were deconvoluted by the XPSPEAK41 program. N_2_ adsorption-desorption isotherms of the samples were performed on a NOVA 2200e analyzer (Boynton Beach, FL, USA) at a temperature of 77 K. The thermal stability of the samples was recorded by a Mettler TGA/DSC3+ thermal analyzer (Schwerzenbach, Switzerland) with a heating rate of 5 °C·min^−1^. Inductively coupled plasma optical emission spectrometer (ICP-OES) determination was executed on an Agilent 700 (Santa Clara, CA, USA) instrument.

## 3. Results and Discussion

Figure 2 illustrates the XRD patterns of Ru@UiO-66 and UiO-66. Obviously, the main diffraction peaks of Ru@UiO-66 at 2*θ* = 7.3°, 8.5° and 25.7° match well with UiO-66, revealing that no significant loss is observed in UiO-66 crystallinity after the introduction of RuCl_3_ in the synthesis of UiO-66. No diffraction peaks corresponding to Ru species can be detected in Ru@UiO-66, which may be ascribed to the low loading amount of Ru species, and/or the encapsulation of the Ru species into the UiO-66 skeleton structure.

FTIR spectra of Ru@UiO-66 and UiO-66 were measured, as illustrated in Figure 3. The FTIR spectra of Ru@UiO-66 and UiO-66 display the same peaks in the region of 4000 to 400 cm^−1^. The broad absorption band in the region of 3700 to 3200 cm^−1^ corresponding to the stretching vibration of O-H, to a large extent, is attributed to the residual solvent and adsorbed water [1,40]. In addition, it can also be induced by the OH from deprotonation of the carboxylate groups. The bands centered at ca. 1700 cm^−1^ and 1400 cm^−1^ correspond to the symmetrical stretching vibrations of the C=O bond in the -COO- group in the framework. The bands at 1506 cm^−1^ and 1581 cm^−1^ are assigned to the C=C stretching vibration of the phenyl ring. These indicate that the main functional groups in the BDC organic linker have been kept for Ru@UiO-66 and UiO-66. Furthermore, the peak at 745 cm^−1^ is consistent with the symmetric vibration peak of O-Zr-O and the symmetric vibration peak of O-Zr-O at 663 cm^−1^ [7,10]. All spectra have weak absorption bands in the region of 600–400 cm^−1^ pertaining to the in-plane and out-of-plane bending vibrations of -COO-. It was proven that Zr, as the coordination center in the organic framework, is formed by bridging with the organic ligand terephthalic acid through the bond of μ_3_-O [41]. However, the vibration peaks related to ruthenium cannot be detected from the FTIR spectra, suggesting that the ruthenium does not exist as a combined state in the UiO-66 skeleton.

The typical SEM images of Ru@UiO-66 and UiO-66 are shown in Figure 4. Ru@UiO-66 exhibits a similar morphology with UiO-66 in spite of the introduction of ruthenium in its skeleton. Furthermore, there are no obvious differences in the grain sizes of Ru@UiO-66 and UiO-66 according to the SEM images. The average grain size distribution for Ru@UiO-66 (202 ± 29 nm) is close to UiO-66 (201 ± 22 nm). Both values demonstrate that in situ addition of the Ru precursor during the fabrication of UiO-66 has no effect on the formation and uniform growth of the grains of Ru@UiO-66.

The XPS spectra of survey, Ru3p, Zr3d, and Cl2p over Ru@UiO-66 are shown in Figure 5. The elements of carbon, oxygen zirconium, and ruthenium are observed clearly in Figure 5a. Among these elements, carbon, oxygen, and zirconium constitute the catalysts. It is noted in Figure 5b that the peak appearing at 462.4 eV is attributed to 3p_3/2_ of Ru^0^ [36]. This reveals that the Ru^3+^ ions from RuCl_3_ have been reduced into metallic Ru^0^ directly, without extra reduction processes. During the synthesis of Ru@UiO-66, BDC and DMF serve as the skeleton of UiO-66 and solvent, respectively, and both of them can provide redundant H species for reducing Ru^3+^ ions. As validation, Ni@UiO-66, Pt@UiO-66, and Pd@UiO-66 have been successfully synthesized by the same method. The peak at 475 eV might be the Auger electron of O KL1L1 and KL1L23, which does not belong to any Ru species [42]. As mentioned in Figure 4, the peaks related to ruthenium are not detected in the FTIR spectrum of Ru@UiO-66, which further suggests that ruthenium existed in the framework of Ru@UiO-66 as a metallic state. The Zr3d_5/2_ and Zr3d_3/2_ in Ru@UiO-66 at 182.7 eV and 185.1 eV are attributed to the Zr^4+^ in O-Zr-O [7], which is also confirmed by the FTIR result (Figure 3). As shown in Figure 5c, no Cl can be detected, suggesting that the Cl^-^ ions have been removed by washing during preparation of Ru@UiO-66.

The HRTEM and mapping measurements were further completed to observe the structure of Ru@UiO-66 as presented in Figure 6. Figure 6a,b clearly display that Ru NPs are regularly encapsulated in the framework of UiO-66 and {101} lattice fringes with an interplanar spacing of 0.204 nm [43]. This is in accordance with the low content of Ru^0^ on surface XPS spectra in Figure 5. The Ru NPs size distribution is presented in Figure 6c, it shows that Ru NPs have a small particle diameter of 1.66 nm and central distribution in the framework of UiO-66. Mapping analysis in random areas in Figure 6d–h reveals that C, O, and Zr elements abundantly exist in Ru@UiO-66, which are the main components of UiO-66 skeleton. In addition, Ru elements are evenly distributed in Ru@UiO-66.

Figure 7 gives the nitrogen adsorption-desorption isotherms and the classical Barrett–Joyner–Halenda (BJH) pore size distribution curves of Ru@UiO-66 and UiO-66. The specific surface area of Ru@UiO-66 is 766.4 m^2^·g^−1^, as calculated by the Brunauer–Emmett–Teller (BET) method, which is slightly lower than that of UiO-66 (996.9 m^2^·g^−1^). The pore diameter of Ru@UiO-66 (3.4 nm) is equal to UiO-66. This indicates that the addition of the precursor RuCl_3_ in the synthesis of Ru@UiO-66 did not affect the growth of the UiO-66 skeleton structure, and the pore size and pore volume of Ru@UiO-66 are similar to that of UiO-66.

To evaluate the thermal stability of Ru@UiO-66 and UiO-66, thermo-gravimetric analysis (TGA) was carried out in air, as shown in Figure 8. The weight loss of the three samples under 100 °C is attributed to desorption of water and residual solvent adsorbed at the surface. The TGA curve of UiO-66 declines rapidly at about 450 °C, suggesting the decomposition of the organic linker in the framework [8,44,45]. In contrast, the decomposition temperature of Ru@UiO-66 (ca. 350 °C) is lower than that of UiO-66 (450 °C), which may result from the presence of the Ru species. In other words, the interaction between -COO^−^ and Zr^4+^ is weakened due to the existence of the Ru species, and the Ru species facilitate the thermal decomposition of the framework at high temperature.

The activity of Ru@UiO-66 and UiO-66 for plasma-assisted catalytic CO_2_ methanation were tested, as illustrated in Figure 9. For comparison, the activity data of Ru/UiO-66 (UiO-66 supported Ru prepared by incipient wetness impregnation method) taken from a previous work [38] was also illustrated in Figure 9. The loading amount of Ru in Ru@UiO-66 and Ru/UiO-66 are close, and the reaction was performed under the same experimental parameters. The CO_2_ conversion of Ru@UiO-66 is 3.6 times higher than UiO-66, and the CH_4_ selectivity over Ru@UiO-66 slightly increases and reaches 95.4% at steady state. Even more interesting is that the products’ selectivities of the two samples are quite different. The primary product of Ru@UiO-66 is CH_4_, and the yield of CH_4_ reaches 68.9%. While the product of UiO-66 was mainly CO, the yield of CH_4_ was less than 3%. Due to the one-step preparation of Ru@UiO-66, the ruthenium in the precursor RuCl_3_ is directly reduced to the metallic Ru and encapsulated in the skeletal structure of the UiO-66. Therefore, the Ru@UiO-66 catalyst with active Ru species exhibits a perfect performance for plasma-assisted catalytic CO_2_ methanation. In addition, the CO_2_ conversion over Ru/UiO-66 increased from 20.4% for pure UiO-66 to 41.3%. However, it is still much lower than that over the Ru@UiO-66 catalyst (72.2%). Moreover, the CH_4_ selectivity over Ru/UiO-66 reaches 86.5%, while it is only ca. 3% for pure UiO-66. In summary, the Ru/UiO-66 possess much higher CO_2_ conversion activity, CH_4_ selectivity, and yield than pure UiO-66 due to the appearance of active Ru species. However, they exhibit poorer performance than the Ru@UiO-66 catalyst, which was mainly attributed to the encapsulation of highly dispersed active Ru species in the UiO-66 skeleton. These indicate that not only the Ru species but also the preparation methods play important roles in plasma-assisted CO_2_ conversion reactions.

The synergy of cold plasma and active Ru species play a vital role in the catalytic activity of CO_2_ methanation. Lee et al. [34] reported that the conversion of 5.369 wt% Ru/Al_2_O_3_ reached 23.2% under the synergy of DBD plasma. This is beyond the conversion of pure plasma and catalysts. Although this is the first report about the co-activation and conversion of CO_2_ by Ru-based catalysts with plasma, the conversion of CO_2_ is unsatisfactory. As for UiO-66, with a strong adsorption ability of CO_2_, it can serve as the carrier of the metal catalyst to effectively improve CO_2_ conversion. Unlike Ru/UiO-66, the active Ru NPs are distributed in the framework of Ru@UiO-66 uniformly to participate in the CO_2_ methanation effectively. The high catalytic activity of Ru@UiO-66 is due to the high dispersion of Ru NPs and the high CO_2_ adsorption ability of the UiO-66 skeleton.

## 4. Conclusions

In summary, Ru@UiO-66 was successfully synthesized via a one-step solvothermal method. The Ru species from the RuCl_3_ precursor was reduced directly and embedded into the framework of UiO-66 during the synthesis of Ru@UiO-66, without an additional reducing process. The crystallinity and structural integrity of UiO-66 were maintained well after encapsulating Ru NPs, and the Ru NPs were uniformly distributed inside the framework of UiO-66. The results of the plasma-assisted CO_2_ methanation indicated that Ru@UiO-66 exhibited an unprecedented catalytic activity with the synergy of plasma. CO_2_ conversion and CH_4_ selectivity over Ru@UiO-66 reached 72.2% and 95.4% under 13.0 W discharge power and 30 mL·min^−1^ gas flow rate ($V_{H_{2}}:V_{CO_{2}}=4:1$). The high performance of Ru@UiO-66 can be ascribed to the synergy of the Ru NPs and cold plasma. The high CO_2_ adsorption ability of UiO-66 is essential for CO_2_ methanation. This work provides a simple method to synthesize high-performance MOF-supported Ru catalysts via a one-step solvothermal method to reduce and encapsulate the Ru NPs into the skeleton at the same time.

## Figures and Tables

**Figure 1 nanomaterials-09-01432-f001:**
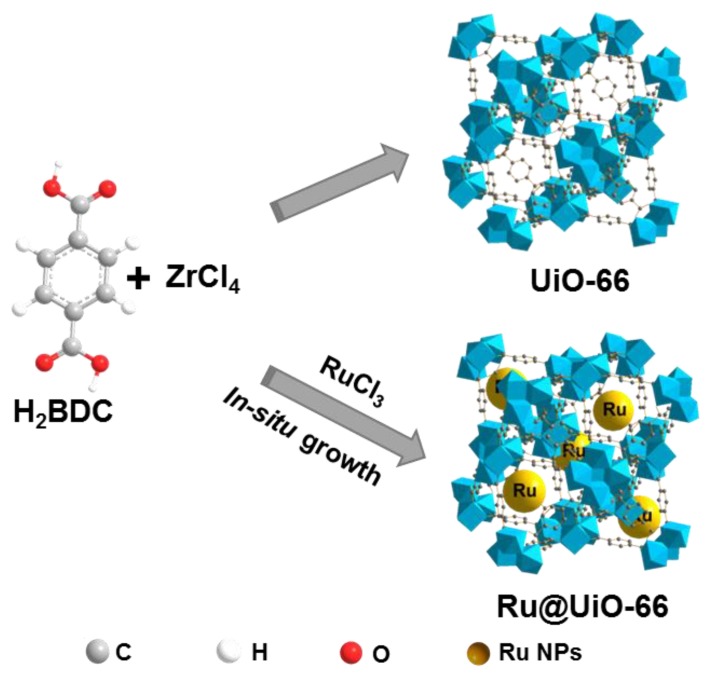
Schematic illustration for the synthesis of UiO-66 and Ru@UiO-66.

**Figure 2 nanomaterials-09-01432-f002:**
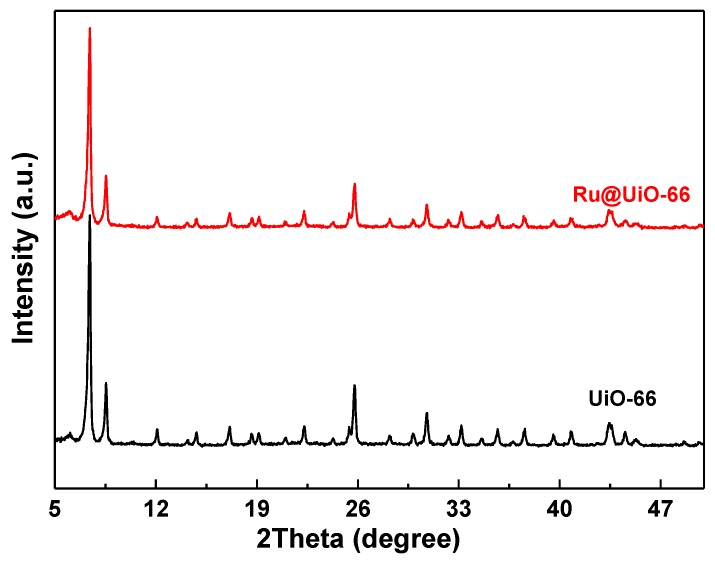
X-ray diffraction (XRD patterns of Ru@UiO-66 and UiO-66.

**Figure 3 nanomaterials-09-01432-f003:**
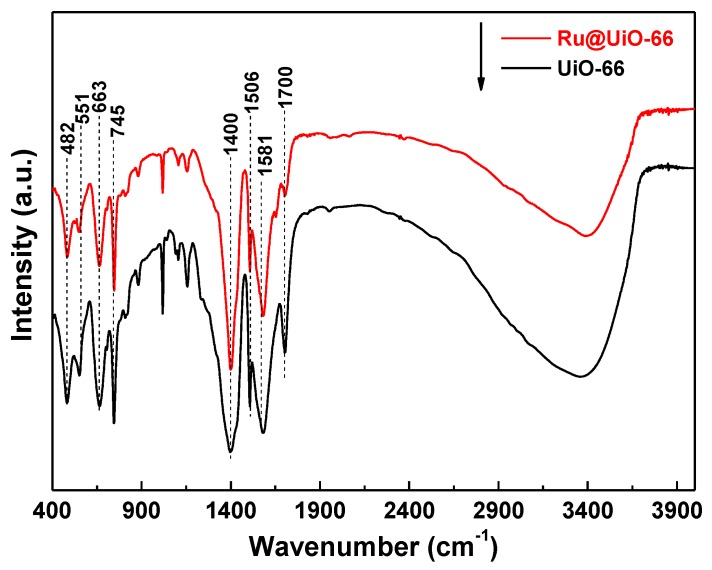
Fourier transform infrared spectroscopy (FTIR) spectra of Ru@UiO-66 and UiO-66.

**Figure 4 nanomaterials-09-01432-f004:**
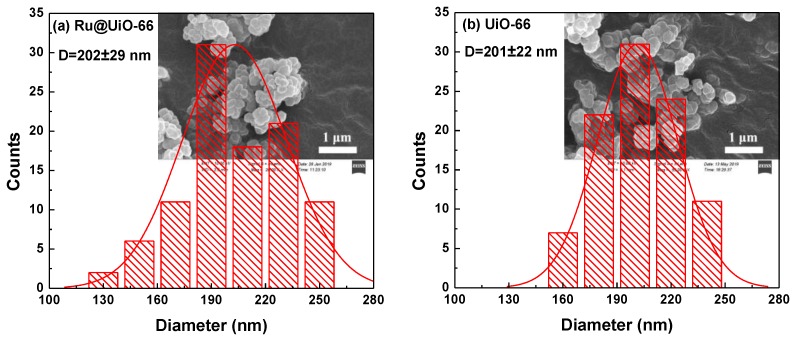
Scanning electron microscopy (SEM) images and the corresponding histograms of the size distribution of (**a**) Ru@UiO-66 and (**b**) UiO-66.

**Figure 5 nanomaterials-09-01432-f005:**
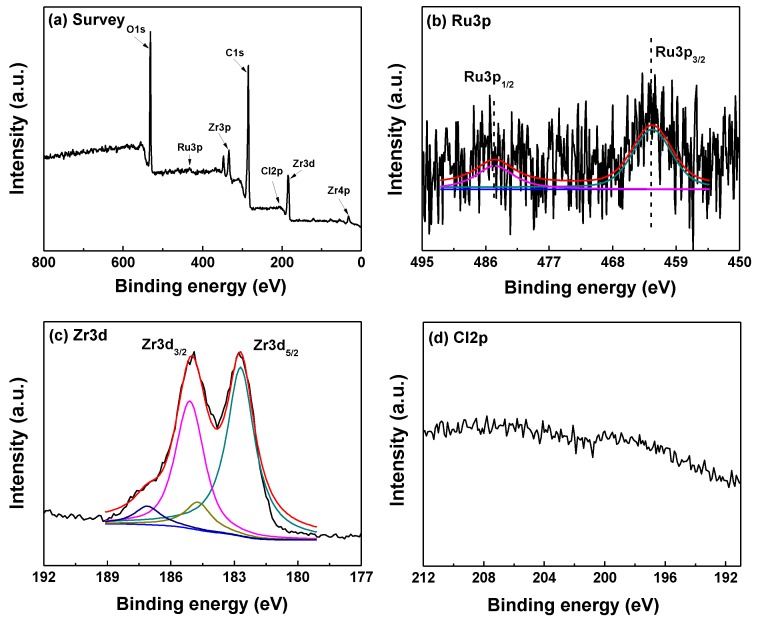
The X-ray photoelectron spectroscopy XPS spectra of (**a**) survey, (**b**) Ru3p, (**c**) Zr3d, and (**d**) Cl2p over Ru@UiO-66 and UiO-66.

**Figure 6 nanomaterials-09-01432-f006:**
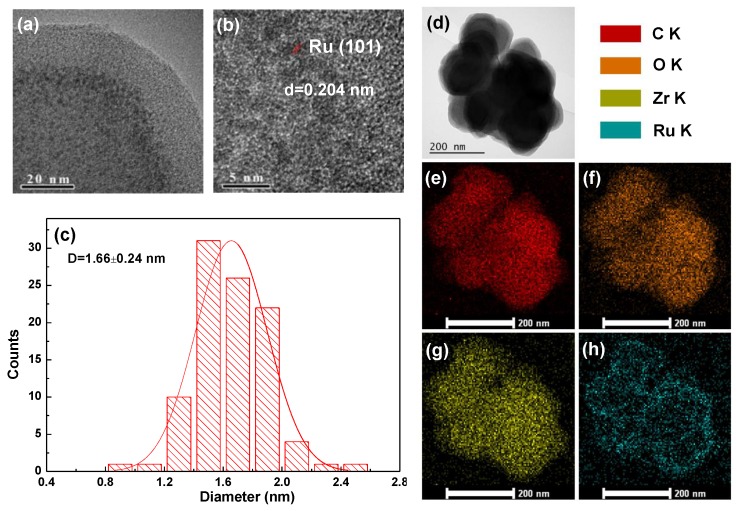
(**a**,**b**) The HRTEM images of Ru@UiO-66. (**c**) The size distribution histogram of Ru NPs. (**d**–**h**) The mapping of C, O, Zr, Ru elements on Ru@UiO-66 catalyst.

**Figure 7 nanomaterials-09-01432-f007:**
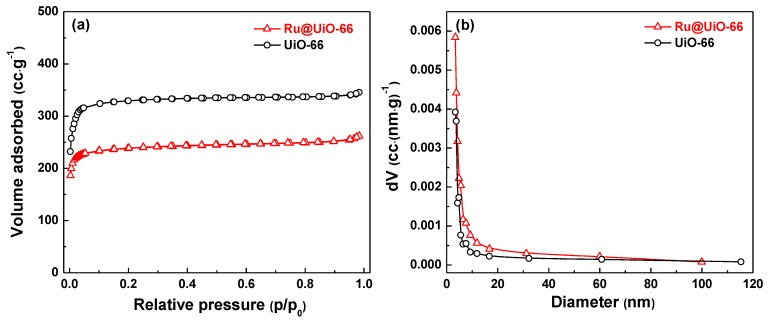
(**a**) Nitrogen adsorption-desorption isotherms and (**b**) the BJH pore size distribution curves of Ru@UiO-66 and UiO-66.

**Figure 8 nanomaterials-09-01432-f008:**
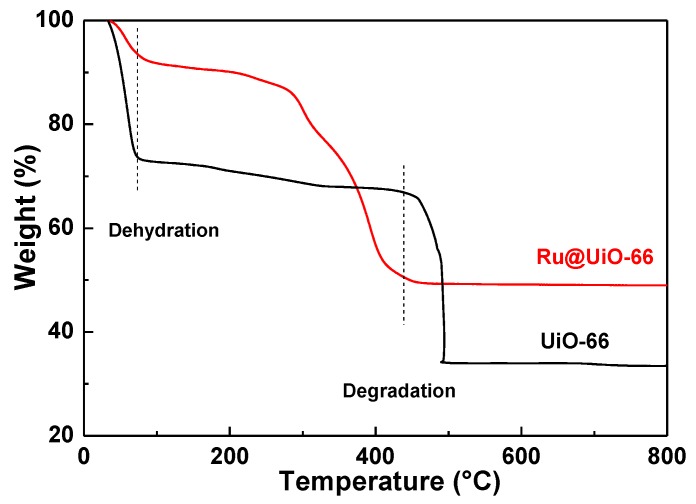
Thermogravimetric analysis (TGA) curves of Ru@UiO-66 and UiO-66.

**Figure 9 nanomaterials-09-01432-f009:**
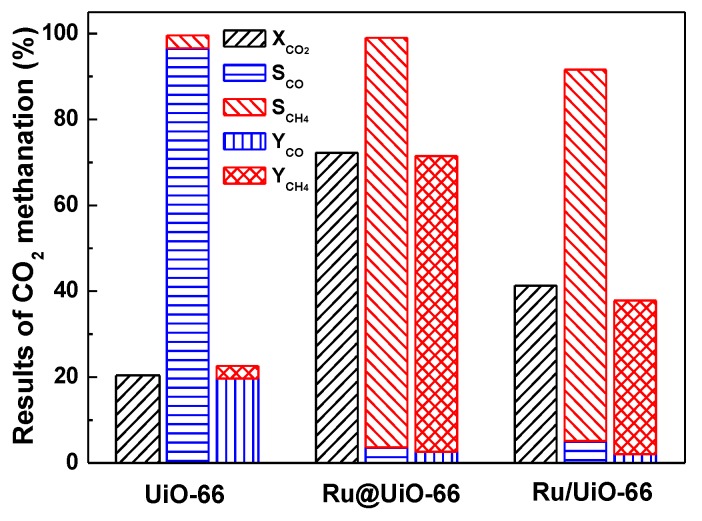
Dielectric barrier discharge (DBD) plasma-assisted CO_2_ conversion over UiO-66, Ru@UiO-66, and Ru/UiO-66, Reproduced from [38], Copyright Hefei Institutes of Physical Science, Chinese Academy of Sciences and IOP Publishing, 2019.

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
