# Peer review of "A Facile Method for Preparing UiO-66 Encapsulated Ru Catalyst and its Application in Plasma-Assisted CO2 Methanation"

_nanomaterials, 2019, doi:10.3390/nano9101432_

Round 1

Reviewer 1 Report

In my past comments, i asked to modify the paper highlighting the difference of Ru@UiO-66 system with commercial Ru/C. Authors should reply to this important point.

I copy here my past comments:

Please explain labels of Fig. 8 (DBD plasma-assisted CO2 conversion over Ru@UiO-66 and UiO-66) Products selectivity of the tested catalysts (Ru@UiO-66 and UiO-66) is not surprising since the active site for CO2 methanation is Ruthenium. For this reason authors should compare the results of Ru@UiO-66 system with some commercial Ru based catalyst (Ru/C for example) or (even better) with an analogous ru-based catalytic system obtained after impregnation of UiO-66. It is not clear to metal loading for Ru@UiO-66.

Finally, the paper is very concise. I therefore suggest to transform in a “Short Communication”.

Reviewer 2 Report

The manuscript is properly revised.

Author Response

Thank you for your positive comments.

Round 2

Reviewer 1 Report

Authors full address raised points. Paper can now be accepted for publication.

This manuscript is a resubmission of an earlier submission. The following is a list of the peer review reports and author responses from that submission.

Round 1

Reviewer 1 Report

1) In Figure 1, authors draw the structure of Ru@UiO-66. Theoretically, authors can calculate the atomic % (or weight %) of Ru of Ru@UiO-66.

Is it well matched with authors' ICP data (Ru contnet is 2.83 wt%.).

Please discuss the proposed structure of of Ru@UiO-66  with relevance to ICP data.

2. Main application of this paper is catalytic property. Authors only represent the catalytic property of  Ru@UiO-66  in terms of conversion (Fig. 8). Authors need to estimate catalytic properties by quantitative method.

It should be presented using TOF value. TOF should be compared to literatures.  

3. I think that Ru@UiO-66  is a new compound. If it is like that, TEM (EDX, HRTEM) and EELS (if possile) should be required.

Reviewer 2 Report

In this contribution it is reported the preparation of Ru NPs encapsulated in UiO-66 metal-organic framework via a solvothermal method.

Ru@UiO-66 catalyst was characterized by XRD, FTIR, TGA, SEM and XPS analysis and its catalytic activity was tested in the plasma-assisted catalytic CO2 methanation. 

The part of the paper related to the synthesis and characterization of Ru@UiO-66 catalyst is quite solid, however, authors should give more details on catalytic tests.

Some points:

Please explain labels of Fig. 8 (DBD plasma-assisted CO2 conversion over Ru@UiO-66 and UiO-66) Products selectivity of the tested catalysts (Ru@UiO-66 and UiO-66) is not surprising since the active site for CO2 methanation is Ruthenium. For this reason authors should compare the results of Ru@UiO-66 system with some commercial Ru based catalyst (Ru/C for example) or (even better) with an analogous ru-based catalytic system obtained after impregnation of UiO-66. It is not clear to metal loading for Ru@UiO-66.

Finally, the paper is very concise. I therefore suggest to transform in a “Short Communication”.